# A Deep View of the Biological Property of Interleukin-33 and Its Dysfunction in the Gut

**DOI:** 10.3390/ijms241713504

**Published:** 2023-08-31

**Authors:** Yi Wang, Chengwei He, Shuzi Xin, Xiaohui Liu, Sitian Zhang, Boya Qiao, Hongwei Shang, Lei Gao, Jingdong Xu

**Affiliations:** 1Department of Clinical Medicine, School of Basic Medical Sciences, Capital Medical University, Beijing 100069, China; m18701692895@163.com (Y.W.); sitianccmu@163.com (S.Z.); qiaoboya2004@163.com (B.Q.); 2Department of Physiology and Pathophysiology, School of Basic Medical Sciences, Capital Medical University, Beijing 100069, China; hcw_1043@foxmail.com (C.H.); xinshuzi@gmail.com (S.X.); m13484363443@163.com (X.L.); 3Experimental Center for Morphological Research Platform, Capital Medical University, Beijing 100069, China; hongwei@ccmu.edu.cn; 4Department of Intelligent Medical Engineering, School of Biomedical Engineering, Capital Medical University, Beijing 100069, China

**Keywords:** IL-33, inflammatory bowel disease, gastrointestinal cancer, nuclear IL-33, biological characteristics

## Abstract

Intestinal diseases have always posed a serious threat to human health, with inflammatory bowel disease (IBD) being one of them. IBD is an autoimmune disease characterized by chronic inflammation, including ulcerative colitis (UC) and Crohn’s disease (CD). The “alarm” cytokine IL-33, which is intimately associated with Th2 immunity, is a highly potent inflammatory factor that is considered to have dual functions—operating as both a pro-inflammatory cytokine and a transcriptional regulator. IL-33 has been shown to play a crucial role in both the onset and development of IBD. Therefore, this review focuses on the pathogenesis of IBD, the major receptor cell types, and the activities of IL-33 in innate and adaptive immunity, as well as its underlying mechanisms and conflicting conclusions in IBD. We have also summarized different medicines targeted to IL-33-associated diseases. Furthermore, we have emphasized the role of IL-33 in gastrointestinal cancer and parasitic infections, giving novel prospective therapeutic utility in the future application of IL-33.

## 1. Introduction

Interleukin-33 (IL-33) is an IL-1 family member expressed in the nucleus of endothelial cells and epithelial cells of barrier tissues. It is released following tissue injury and cell death. Since its discovery, IL-33 has been implicated in the onset and progression of asthma [1], cardiovascular diseases, kidney injury [2,3], and allergic diseases [4], as well as in organ fibrosis [5,6]. IL-33 has also recently been associated with Alzheimer’s disease (AD) and rheumatic heart disease [7,8,9]. In addition, IL-33 plays a crucial role in IBD, a chronic intestinal inflammation with no cause, which has a high global incidence rate [10]. Due to lower exposure to infectious microorganisms in developed countries—which limits the maturation of the intestinal immune system—IBD is far more prevalent in developed countries than in rising industrialized countries like China [11,12,13]. IBD includes Crohn’s disease (CD) and ulcerative colitis (UC), which are distinguished by symptoms such as diarrhea, bloody stool, abdominal pain, and weight loss. Patients with CD experience Th1-type immunity, with activation of Th17 and Th1 cells and insufficient Treg cells [14]. CD can occur from the oral cavity to the rectum [15]. Patients with UC, on the other hand, demonstrate Th2-type immunity, with a more pronounced increase in IL-5 and IL-13 [16]. UC causes ulcers in the mucous membrane, which are generally restricted to the colon [15,17].

It is widely acknowledged that genetics, the environment, gut flora, and immunological factors all have an impact on the development of IBD [18,19]. IL-33 plays a crucial role in the initiation and amplification of type 2 immune responses and allergic inflammation. However, research on the relationship between IL-33 and intestinal disorders is sparse. The aim of this review is to summarize the role of IL-33 in the gut based on available research findings while also highlighting current research limitations in order to present an essential theoretical foundation for future clinical practice.

## 2. Biological Characteristics of IL-33

IL-33, a nuclear factor expressed in human high endothelial venules [20], belongs to the IL-1 family of cytokines and is expressed in the nucleus of endothelial cells and epithelial cells of barrier tissues. IL-33, released upon tissue or cellular damage, is one of the ligands for ST2 and is known as an “alarmin” cytokine. It serves as a potent driver for the production of Th2 cytokines and is involved in allergic reactions and Th2 immune responses in autoimmune disorders [21].

Full-length IL-33 (flIL-33) is a precursor protein with a molecular weight of 30 kDa and lacking an N-terminal signal peptide. Its encoding gene is located on human chromosome 9p24.1 and mouse chromosome 19qC1 [20,22]. IL-33 has double roles: as a cytokine and as a transcriptional regulator. It has three main domains: an N-terminal nuclear localization domain, a central protease-sensing domain, and a C-terminal cytokine domain. IL-33 is released passively during cell necrosis, activation via non-cell death ATP pathways, or tissue damage. The protease-sensing domain of IL-33 can interact with inflammatory proteases from immune cells such as neutrophils (including tissue protease G and elastase) and mast cells (including chymase and tryptase), leading to the degradation of flIL-33 into a shorter mature form known as mature IL-33 (18 kDa), which retains the C-terminal cytokine domain and exhibits 10–30 times higher activity compared with flIL-33 [23,24,25].

Once flIL-33 is released extracellularly and cleaved into a mature form, it acts as a cytokine. However, intracellular flIL-33 can bind to chromatin and operate as a transcriptional regulatory nuclear factor known as nuclear IL-33 [26]. IL-33 is primarily expressed by non-hematopoietic cells, particularly intestinal epithelial cells, as well as myofibroblasts and endothelial cells [27]. IL-33 is also expressed in mast cells, Th2 cells, eosinophils, natural killer (NK) cells, and other cell types [28,29]. Other studies, however, have corroborated that during acute inflammation, IL-33 is mainly expressed by colonic fibroblasts and is independent of the IL-1β pathway [30]. IL-33 can activate Th2 cells, mast cells, eosinophils, Treg cells, ILC2, NK cells, Th1 cells, dendritic cells, and macrophages [31,32]. In contrast to other members of the IL-1 family (such as IL-1 and IL-18, which primarily promote type 1 immune responses), IL-33 generally promotes type 2 immune responses [22] and is involved in allergic disorders, parasitic infections, and inflammation [33]. Therefore, IL-33 inhibitors might serve as a potential breakthrough in relieving allergic diseases.

### 2.1. The IL-33 Release Mechanism

Unlike other members of the IL-1 family, such as IL-18 and IL-1β, IL-33 is not released via the endoplasmic reticulum or Golgi apparatus. The most common mode of release is its passive release following infection or tissue damage-induced cell death [34,35,36]. Like IL-1β and IL-18, IL-33 was initially thought to require cleavage by caspase-1 to become biologically active [37]. However, further studies revealed that flIL-33 is biologically active and loses activity upon cleavage by activated caspase-3 and caspase-7 [29,38]. This additionally clarifies why IL-33 is not released during cellular apoptosis, thus avoiding inflammation. Other IL-33 release routes—aside from tissue necrosis—are presently being investigated. Recent research revealed that binding of IL-33 to ST2 can promote the translocation of perforin-2 from the cytoplasm to the cell membrane, forming pores and providing a conduit for IL-33 release from intracellular to extracellular location (Figure 1B) [39]. However, the mechanism through which IL-33 facilitates the translocation of perforin-2 is unidentified, and further research into this process might reveal novel understanding and strategies for suppressing IL-33 release. ATP release from damaged intestinal epithelial cells can considerably enhance IL-33 release [40]. Mechanical stress can also cause IL-33 release [41].

In addition, O-GlcNAc transferase (OGT) modifies STAT6 into STAT6 O-GlcNAc, which enhances Gsdmc2–4 and the formation of Gsdmc pores on the cell membrane, allowing IL-33 secretion (Figure 1A) [13]. The authors suggested that Gsdmc may not be necessary for acute injury induced by DSS, possibly because epithelial damage caused by DSS is a key factor that leads to the release of IL-33. However, the principal mechanisms of IL-33 release and its activities can shift under different pathological and physiological situations, and the expression of perforin-2 and Gsdmc may differ between cell types. As a result, the significance of the Gsdmc pathway in various clinical and physiological states remains unknown.

### 2.2. The Transcriptional Regulatory Role of IL-33 in the Cell Nucleus

IL-33 possesses a DNA binding domain (HTH) in its N-terminal nuclear localization region that facilitates its interaction with chromatin [21]. This interaction is mediated across protein–protein interactions via the CBM region rather than via protein–DNA interactions [42]. Through a nuclear localization sequence, IL-33 is synthesized in the cytoplasm and transferred to the nucleus for storage. The specific role of nuclear IL-33 is currently unclear, with some studies suggesting its role as a transcriptional regulator [43]. Waddel’s experiments revealed shifts in cytokine expression in the tissues of IL-33^−/−^ and ST2^−/−^ mice [44]. Given that ST2 is currently the sole known receptor for IL-33, it can be assumed that nuclear IL-33 has undiscovered roles.

Nuclear IL-33 may play a role in the transcriptional activation and transcriptional repression of specific genes in various immunological settings. Evidence from Ni’s experiment revealed that nuclear IL-33 can undergo polyubiquitination modification in HEK293T cells, and USP17 can deubiquitinate IL-33 to stabilize its activity and expression as well as regulate the gene expression of IL-13, to some extent [21]. In addition, nuclear IL-33 can bind to the p65 promoter, resulting in the overexpression of cell adhesion molecules ICAM-1 and VCAM-1, which are involved in the regulation of cell integrity and inflammation [45]. Nuclear IL-33 may also have an inverse impact on NF-κB during homeostasis, limiting pro-inflammatory effects [46]. Conversely, endogenous IL-33 in the tumor microenvironment has been postulated to inhibit NF-κB expression, hence reducing the induction of “fragile” Foxp3+ Tregs and promoting the production of IFN-γ [47].

Recent studies show that nuclear IL-33 interacts with RUNX2, inhibiting the expression of Smad6 and relieving its inhibition on the TGF-β/Smad pathway, thus promoting epithelial cell proliferation and potentially facilitating the transition from chronic skin inflammation to cancer [48]. Using a mouse model of Sendai virus lung infection, Wu and his colleagues revealed that nuclear IL-33 can promote the expansion of alveolar basal cells, aiding in early improvement of lung alveolar barrier function [49]. Nuclear IL-33 can also regulate the expression of chemokines such as CCL5 [50].

Studies have also shown that deleting the nuclear sequence of IL-33 increases the extracellular release of IL-33, resulting in systemic inflammation in mice [51]. Notably, nuclear IL-33 interacts dynamically with chromatin, limiting the extracellular release of IL-33 during cell death by activating the IL-33/ST2 pathway with histones and inducing cytokine production, particularly CCL2. Conversely, other studies suggest that the overexpression of IL-33 during steady-state has neither an effect on gene expression [35] nor possesses transcriptional regulatory effects [43].

### 2.3. IL-33 Exerts Its Function in Innate Immunity

During endothelial cell damage and necrosis, intracellular IL-33 is released into the tissue, where it acts as a cytokine [29]. Injecting recombinant IL-33 (rIL-33) into normal mice enhanced the proliferation of intestinal goblet cells as well as the infiltration of eosinophils and neutrophils [22].

Paneth cells, located at the base of the small intestinal crypts, can secrete various antimicrobial peptides that have been linked to the prevention, treatment, and maintenance of IBD and the immune system’s tolerance to the gut flora [52,53]. Not only does IL-33 induce the production of REG3γ via mTOR, STAT3, and ERK1/2 signaling pathways, regulating gut flora and suppressing bacterial (Figure 2), it also promotes its expression in intestinal epithelial cells (IECs) [54] and promotes the differentiation of IECs into goblet cells and aids wound healing, alleviating inflammation (Figure 2) by inducing the polarization of M2 macrophages [55]. Aside from its indirect effects, IL-33 can also directly act on intestinal epithelial cells to increase the level of miR-320, which promotes epithelial cell repair and proliferation [56]. Furthermore, miR-320 is now considered a potential biomarker of IBD, as its expression profile varies in different intestinal diseases and phases of the disease and is linked to IBD diagnosis and prognosis [57].

#### 2.3.1. Double-Edged Sword Effect of IL-33 and ILC2 on Intestinal Epithelium

Innate lymphoid cells (ILCs), located on mucosal surfaces, can release cytokines such as IL-5 and IL-13, which aid in the immune response during intestinal worm infections. ILC2, whose development and maturation are closely associated with GATA3 [58], is classified into two subtypes (iilc2 and nilc2) [59] with a crucial role in maintaining the integrity of the intestinal epithelium and may have a predominant effect during the early stages of colitis [60]. ILC2 proliferates rapidly in response to IL-33 activation and produces cytokines such as IL-13, IL-5, and IL-9, playing a crucial role as a major source of type 2 cytokines during early infection [59,61]. IL-13 can induce proliferation and differentiation of goblet cells in the lung and intestinal epithelium (Figure 2), which are not observed in IL-13-deficient mice [62,63]. This was corroborated using RAG2^−/−^γc^−/−^ and RAG2^−/−^ mice [64].

Recent experiments by Schumacher et al. confirmed that IL-33 does not directly stimulate goblet cell proliferation and MUC2 secretion, but instead relies on the production of IL-13 (Figure 2) [63,65], consistent with previous findings. In addition to its effects via IL-13, IL-33 can induce ILC2 to produce AREG, which, upon binding to epithelial cell EGFR, promotes epithelial barrier repair and mucin production (Figure 2) [66].

Relevant experiments have revealed that ILC2s are associated with fibrosis in the liver, kidney, lungs, and intestines [67,68,69,70], indicating their pathogenicity in chronic inflammation. Moreover, ILC2s may acquire the ability to produce IFN-β, which can contribute to intestinal inflammation [71]. Interestingly, recent studies have revealed that the induction of type 2 cytokine production in ILC2s by IL-33 can be inhibited by peroxisome proliferator-activated receptor gamma (PPARγ) activation [72]. This provides insights into potential strategies for treating the pathological effects that may arise from the interaction between IL-33 and ILC2s. Additionally, gastric mucosal commensal bacteria and pathogenic bacteria such as *Helicobacter pylori* induce the production of IL-7 and IL-33, which in turn activate ILC2s to proliferate and secrete IL-5, leading to B cell secretion of IgA that wraps symbiotic and pathogenic bacteria together, thus providing protective effects [73].

The above facts highlight that IL-33 plays a critical role in inducing ILC2s for the protection of intestinal epithelium via different pathways. Although an investigation of ILC2 regulation mechanisms and IL-33 pathways is required, IL-33 and related LC provide a fresh avenue for clinical application.

#### 2.3.2. IL-33 Induces Eosinophil Recruitment

Numerous studies have confirmed the importance of eosinophils in allergic reactions, with IL-5 being a crucial component. CCL11 and CCL24 are also implicated in eosinophil recruitment and activation [74]. Aggregation of eosinophils is often observed in allergic airways, esophagitis, and IBD. Previous studies widely regarded eosinophils as pro-inflammatory cells owing to the fact that their aggregation promotes inflammation [75]. However, they may assist in the preservation of the epithelial barrier and regulation of local immune responses [76]. Evidence obtained by Sugawara demonstrated that eosinophils in the small intestines may release IL-1Ra, which inhibits Th17 cells via downregulation of IL-1β, IL-6, and IL-17A [77]. Aside from the proliferation and differentiation of eosinophils, IL-33 upregulation in the ileum during inflammation promotes the production of IL-5, IL-8, IL-13, CCL1, and CCL5 (Figure 3A), thereby facilitating the infiltration of eosinophils into the intestinal mucosa [74,78]. Moreover, Gurtner’s experiments using a DSS-induced colitis model demonstrated that IL-33 and IFN-γ synergistically promote an increase in the A-Eos subset (with antimicrobial and cytotoxic properties (Figure 2)) in colonic eosinophils, hence minimizing colitis aggravation [76]. We speculate that these differences may be due to the different immunological properties of the colitis mouse model, which also reflects the diversity and complexity of the actions of IL-33.

### 2.4. The Role of IL-33 in Adaptive Immunity

In addition to its effects on innate immune cells, IL-33 can also activate adaptive immunity, particularly Th2-type immune responses, and plays an immunoregulatory role. Here, we summarize the interaction between IL-33 and Treg cells and show how IL-33 regulates Th2-type immunity.

#### 2.4.1. IL-33 Regulates the Proliferation of Treg Cells

Tregs play a crucial role in the immune system by regulating or suppressing other cells to maintain immune homeostasis and prevent excessive immune responses. Their deficiency can lead to autoimmune diseases and the formation of tumor microenvironments [47]. Foxp3^+^ GATA3^+^ ST2^+^ Treg cells are the primary target cells of IL-33, which directly acts on ST2+ Foxp3+ Tregs and stimulate IL-2 production by CD11c+ dendritic cells; these in turn facilitate the expansion of Foxp3+ Tregs. Additionally, IL-33 can activate the NF-κB/p38 signaling pathways to selectively expand ST2^+^ Treg cells [5] as well as increase the percentage of Foxp3^+^ Tregs only in the presence of TGF-β1 (Figure 3B). Furthermore, IL-33 acting on Treg cells can directly promote the expression of ST2, forming a positive feedback loop for the expansion of Foxp3^+^ Treg cells [79]. IL-2 plays a crucial role in the growth and development of Treg cells, and IL-33 may enhance the sensitivity of Treg cells to IL-2 by directly binding to ST2. Additionally, IL-33 can stimulate DC cells to secrete IL-2, thereby increasing IL-2 signaling and promoting the proliferation and survival of Treg cells, hence enhancing their immunosuppressive function. IL-2 plays a crucial role in the growth and development of Treg cells, and IL-33 may enhance the sensitivity of Treg cells to IL-2 by directly binding to ST2 [80]. Recent research indicates that the selective expansion of Foxp3^+^ GATA3^+^ ST2^+^ Treg cells is driven by IL-33 derived from DC cells rather than IECs [39]. In a mouse model of CRC lacking ST2, an increased number of colonic Treg cells is observed without changes in Foxp3 and GATA3 expression, indicating increased infiltration without induction of differentiation [81]. This phenomenon may be attributed to the loss of IL-33-mediated Treg cell-promoting effect due to ST2 deficiency. IL-33 signaling has been found to alter the diversity, phenotype, and function of Tregs, potentially contributing to immune suppression and tumor progression. An important point to mention is that the regulation of Treg cells by IL-33 signaling varies depending on the specific immune microenvironment.

In summary, IL-33 plays a role in the regulation and proliferation of Treg cells. It promotes the expansion and stability of Tregs, enhancing their suppressive function. IL-33 signaling can contribute to tumor immunoevasion by promoting Treg cell proliferation. Further research is needed to fully understand the mechanisms and specific roles of IL-33 in the regulation of Treg cells.

#### 2.4.2. IL-33 Induces Th2 Immune Responses

IL-33 plays a crucial role in regulating Th2 immune response, with Th2 cells being the main cell type expressing ST2. The balance between Th2, Th1, and Th17 cells is essential in the development of colitis [12]. IL-33 predominantly regulates the Th2 immune response, as well as recruits Th2 cells, induces their polarization, and promotes the production of Th2 cytokines such as IL-4, IL-5, IL-10, and IL-13 [5,22]. Consistently, the use of rIL-33 can promote the shift from Th1 to Th2 immune response [14], as well as IL-13 production, depending on the formation of a signaling complex between ST2 and EGFR induced by AREG [82]. In Peter’s experiment, the presence of IL-33 exacerbated DSS-induced colitis in mice, while genetically knocking out IL-4 abolished this effect, suggesting that IL-33 primarily worsens colitis through the Th2 immune response [83]. Junfeng Zhu’s study also demonstrated that IL-33 induces the production of the Th2 cytokines IL-5 and IL-13 while inhibiting Th1 and Th17 immune responses through its action on ILC2 [84]. It is worth noting that IL-33 can also indirectly induce the polarization of Th2 through the activation of DC cells, which is associated with the transcription factors GATA3, GATA5, and GATA6; this promotes the expression of ST2 on Th2 cells [85,86], forming a positive feedback loop in IL-33 signaling. These findings illustrate the wide range of methods through which IL-33 controls the Th2 immune response. Although numerous studies demonstrate that IL-33 is the dominant inducer of the Th2 immune response, the particular mechanisms underlying this induction remain unknown and require further exploration.

### 2.5. The “Sentry” Role of IL-33

IL-33 has been described as an “alarmin” cytokine that acts in response to various types of tissue damage and plays a role in the regulation of both innate and adaptive immune responses. It can be released from cells in response to various forms of cellular damage or stress, including injury, stress, or inflammatory cell death, and its secretion is widely regarded as an “alarm” signal that initiates inflammatory responses. With the progress in IL-33 research, IL-33 released by the intestinal epithelium has been shown to act as an immunoadjuvant for enhancing antigen-specific immune responses and type 2 immunity [13]. IL-33 is enzymatically cleaved and inactivated during normal cell apoptosis, but when tissue and epithelial damage occur, cell necrosis leads to the release of intact IL-33, which exerts its cytokine effects extracellularly. Due to this characteristic, IL-33, like IL-25 and TSLP, can serve as an alarm signal indicating tissue damage [87]. This reflects the important role of IL-33, to some extent highlighting its “sentry” role. Interestingly, some immune cells such as Treg and ILC2 can slowly, and at low levels, release IL-33 without triggering an immune response [31]. This may partially reflect the role of IL-33 in maintaining immune system homeostasis beyond its alarm signaling function. Further research is needed to fully understand the mechanism and regulation of IL-33 as an alarmin cytokine.

### 2.6. The Role of IL-33 in the Gut

IL-33, functioning as a cytokine in the intestinal environment, regulates the intestinal immune response. Its synthesis and release inside the intestinal mucosa activate different immune cell populations, amplify localized inflammatory responses, and play a pivotal role in infection, immunological tolerance, and immune response. IL-33, on the other hand, regulates the activation and proliferation of regulatory T (Treg) cells, while inhibiting the differentiation of T helper 17 (Th17) cells, hence maintaining the stability of the intestinal mucosa. Concurrently, IL-33—whether directly or indirectly—participates in the repair of damaged epithelial layers, the maintenance of intestinal barrier function, and the prevention of inflammation and diseases caused by weak intestinal barrier integrity.

Overall, IL-33 has a wide range of roles in the gut, including immunological modulation, allergy reactions, immune homeostasis, tumor immunity, and intestinal barrier fortification. A comprehensive understanding of IL-33′s involvement in these processes will elucidate its relevance in both intestinal health and disease settings.

## 3. Classification and Expression of ST2

ST2 (growth stimulation gene 2) is a gene that encodes the receptor for IL-33 and is located on human chromosome 2q12 and mouse chromosome 1 within the IL1RL1 gene locus. ST2 has four subtypes, but particular attention has been given to the transmembrane receptor ST2L (referred to as ST2) and the soluble form sST2. ST2 was initially shown to be expressed on Th2 cells, but recent studies have revealed its predominant expression on innate immune cells such as mast cells, ILC2, macrophages, dendritic cells, eosinophils, and natural killer cells, as well as on CD4^+^ T cells, CD8^+^ T cells, and Treg cells [6,33,85,88].

### 3.1. ST2L

ST2 is present in group 2 innate lymphoid cells (ILC2), helper and regulatory T cells, mast cells, basophils, eosinophils, NKT cells, and NK cells. ST2 recruits IL1RAcP to form a signaling complex upon binding to IL-33 [22,89,90]. The dimerization of the TIR domains of ST2 and IL1RAcP receptors leads to the binding of MyD88, which then associates with downstream protein kinases IRAK and TRAF6, activating TRAF6 and activating downstream NF-κB and MAPK pathways [5]. Under different immunological states, ST2 can also form complexes with other cell receptors, such as receptor tyrosine kinase c-Kit and epidermal growth factor receptor (EGFR), to play immunomodulatory roles [31].

### 3.2. sST2

sST2, the soluble derivative of ST2, might attach to IL-33, sharing the same extracellular domain as the membrane-bound ST2L but without the transmembrane and intracellular domains [4]. Increased expression of total ST2 in the colonic epithelium has been observed during inflammation in patients with UC. However, unlike sST2, the expression of ST2L in the epithelium is diminished [91]. This leads to an increase in the sST2/ST2L ratio, which may be an underlying mechanism for downregulating the pro-inflammatory effects of IL-33 during inflammation. Injecting sST2 into CBA/J mice infected with Amebic trophozoites can inhibit the protective effects of IL-33 in amebiasis [64]. sST2 levels in the serum of patients with IBD are considerably elevated [28,55,92], particularly in patients with UC, and the levels are directly correlated with the severity of inflammation, indicating that sST2 may be used as a potential biomarker of IBD [92]. sST2 is also considered an independent predictive factor for heart failure [93,94,95] and a possible marker for early identification of abdominal aortic aneurysm (AAA) and acute pancreatitis [96,97].

## 4. Regulation of the IL-33/ST2 Axis

The IL-33/ST2 axis is a complex signaling pathway that regulates immune responses and has been implicated in the pathogenesis of several diseases, including IBD. The IL-33/ST2 axis controls epithelial niche regeneration by activating a large multi-cellular circuit, including monocyte differentiation into competent repairing AAMs and ILC2-mediated AAM activation. It can also simultaneously regulate the activation and inhibition of various immune cells, participating in precise modulation within different immune microenvironments. Therefore, the regulation of the IL-33/ST2 axis is essential.

It is widely recognized that SIGIRR (also known as TIR8) is a negative regulatory factor for pro-inflammatory signaling. Furthermore, under stimulation of IL-33, SIGIRR can bind to ST2, thereby inhibiting the IL-33/ST2 axis [98]. sST2 is the soluble form of ST2 that can bind specifically to IL-33 extracellularly, competitively inhibiting the IL-33/ST2 axis. Indeed, consistent with this, the use of sST2 inhibitors can enhance the effects mediated by IL-33 [99]. This phenomenon is well illustrated in airway inflammation following exposure to allergens [4]. A recent study found that diphenyl difluoro ketone EF24 (a curcumin analog) can reduce the level of binding between ST2L and MyD88 and promote the expression of sST2. The combined effects of both may negatively regulate the pro-inflammatory signaling of IL-33 [100]. Additionally, MCPT4 released by mast cells can cleave IL-33 extracellularly, rendering it inactive [101]. Neutrophil-secreted proteases have been shown to cleave various cytokines and chemokines and exhibit strong proteolytic activity against IL-33 [25]. Protease 3, elastase, and cathepsin can inactivate IL-33, and the presence of protease 3 can inhibit the activating effects of elastase and cathepsin on IL-33 [24]. The IL-33/ST2 axis is also regulated by the degradation of ST2L. Studies have shown that GSK3β can phosphorylate ST2L, promoting its ubiquitination and degradation by FBXL19. On the other hand, USP38 can inhibit TRAF, inducing ST2L degradation via the autophagolysosomal pathway [102,103]. Furthermore, IL-33 can restrict its release by intracellular chromatin binding, while being inactivated through oxidation in the extracellular environment, thereby reducing inflammatory responses [35,104].

In conclusion, the regulation of the IL-33/ST2 axis is complex and still not fully understood. Studying the regulatory mechanisms underlying the IL-33/ST2 axis is of significant clinical importance in the treatment of IL-33-related diseases. Further research is needed to fully understand the mechanisms of action of the IL-33/ST2 axis in IBD and to develop targeted interventions.

## 5. IBD Pathogenesis Is Complicated by Cytokines and Multiple Aspects

The pathogenesis of IBD, including CD and UC, involves the presence of pathogenic factors such as abnormal gut microbiota, immune response dysregulation, environmental changes, and gene variations. Recent studies have reported an increasing spectrum of human monogenic diseases that can present with IBD-like intestinal inflammation. A substantial proportion of patients with these genetic defects present with very early onset intestinal inflammation. Research has shown that first-degree relatives such as parents, children, or siblings of individuals with IBD have a 5-fold increased risk of developing the disease. The genetic risk is greater for Crohn’s disease than for ulcerative colitis [10,105]. Studies on Crohn’s disease have consistently implicated NOD2 as the most important gene associated with the disease [18,105] and NOD2 gene mutations have been implicated in disease severity. Currently, there are no specific targeted therapies available that directly modulate NOD2 function for the treatment of CD.

IBD is characterized by a reduction in intestinal microbial diversity [106,107]. Dysbiosis of the gut microbiota, which results in the imbalance between beneficial and harmful bacterial taxa, results in damage to the intestinal microbial barrier [12,60,108]. The mucosal barrier, including antimicrobial peptides and the mucus layer, plays a crucial role in defending against pathogens and maintaining gut homeostasis. Dysregulation of the mucosal defense structures can contribute to the development of IBD. Butyrate is a short-chain fatty acid produced by certain bacteria in the gut, e.g., *Faecalibacterium prausnitzii* and *Roseburia* spp. These bacteria are known for their anti-inflammatory properties, and their quantities are lower in patients with IBD. *Bifidobacterium* and certain strains of *Lactobacillus* have anti-inflammatory effects and can help restore the integrity of the intestinal barrier [109,110].

Numerous studies have highlighted that IL-33 plays a complex role in the regulation of the gut microbiota and the pathogenesis of IBD. The IL-33/ST2 axis interacts with important components of the intestinal tissue, including epithelial cells, gut microbiome, pathogenic bacteria, and immune cells. Current research indicates that *Blautia*, *Faecalibacterium*, and *Ruminococcus* are the microbial groups that are most strongly associated with IBD, and patients with IBD exhibit distinct differences in microbial composition, as well as functional and metabolic products, compared with healthy individuals [111].

From the above facts, we can conclude that IBD etiology is complicated and involves a number of variables, including aberrant gut microbiota, dysregulated immune responses, environmental alterations, and gene variations. Further research is required to completely understand the processes underlying the development of IBD and to design tailored therapies.

## 6. The Role of IL-33 in IBD

IL-33 is a cytokine that has emerged as a critical regulator in several autoimmune and inflammatory disorders, including IBD. However, the role of IL-33 in IBD is not yet clearly defined, and controversial data have been obtained regarding whether serum IL-33 levels correlate with IBD activity; current research suggests that IL-33 has a dual role in IBD.

### 6.1. Evidence of the Role of IL-33 in IBD

IL-33 plays a complex role in the regulation of gut microbiota and the pathogenesis of IBD. However, IL-33 is also involved in the inflammatory response in IBD and can exacerbate the disease. IL-33 promotes IgA production to maintain gut microbial homeostasis and restrain colitis and tumorigenesis. IL-33 can indirectly alter the microbiota to protect against colitis through the promotion of IgA production [73,112].

Human IBD samples show upregulated IL-33 or fIl-33 expression in epithelial and crypt cells during inflammation caused by UC [91] and CD, as well as in the lamina propria [27,28]. Additionally, sST2 levels are elevated, and patients with UC exhibit an increased IL-33/sST2 ratio. In vitro stimulation of PBMCs from patients with CD using gliadin peptides leads to increased expression of IL-33 and ST2, suggesting that the IL-33/ST2 axis may play a crucial role in CD and the direct interaction between PBMCs and peptides is associated with the IL-33/ST2 axis [28].

Sedhom’s experiments using DSS and TNBS-induced colitis models in ST2^−/−^ mice (C57BL/6J) demonstrated that IL-33 has a detrimental effect on intestinal epithelial cells, increasing their permeability and bacterial translocation and leading to more severe intestinal inflammation (Figure 3A) [27]. This was further confirmed using a mouse model infected with *C. rodentium* [113]. Regardless of the debate, utilizing a TNBS-induced colitis mouse model (BALB/c) demonstrated that IL-33 may accelerate the transition from Th1 to Th2 response, ultimately relieving Th1-type inflammation [14]. Using an OXA-induced colitis mouse model (C57BL/6J), Waddell showed that IL-33 can protect intestinal goblet cells (Figure 3B) [44]. The protective role of IL-33 in IBD was also observed in the DSS-induced colitis mouse model (BALB/c) [114,115].

### 6.2. Possible Mechanism of Dual Action of IL-33 in IBD

The contrasting results may be attributed to different experimental approaches. Sedhom utilized ST2^−/−^ mice to block the ST2/IL-33 pathway; on the other hand, Duan administered rIL-33 and IL-33 antibodies [14]. ST2^−/−^ mice may have a different intestinal microenvironment compared with normal mice, and the long-term absence of ST2 could trigger compensatory mechanisms. One of the limitations of global knockout mice is that other immune response pathways involving ST2, which have not been fully studied, could impact the mice and lead to atypical or unexpected results. Additionally, the use of different mouse strains may contribute to disparate outcomes, as there could be genetic variations between the strains. Apart from genetic factors, differences in immune microenvironments can also contribute to varying sensitivities to different diseases.

The dual role of IL-33 in IBD may also be associated with the composition of the gut microbiota. Ordas et al. suggest that the disruption of gut microbial diversity and metabolism could be the starting point of inflammation, as susceptibility to IBD varies in different socioeconomic regions [12]. Additionally, compared with SPF-SAMP mice, GF-SAMP mice do not exhibit significant eosinophil infiltration and Th2 responses [74], and other mouse models of colitis do not display significant inflammation under germ-free conditions [105]. These findings suggest that variations in the types and proportions of gut microbiota can lead to different inflammatory responses. However, the association between IL-33 and the gut microbiota and immune system in IBD remains unexplored, and the dual role of IL-33 may be linked to the composition of the gut microbiota. This hypothesis requires further research for confirmation.

The role of IL-33 may vary significantly at different stages due to changes in the intestinal immune microenvironment during the progression of inflammation. Loetuso’s experiments indicate that using anti-IL-33 molecules during the acute stage of inflammation reduces the severity of inflammation. However, the absence of IL-33 during the restoration stage can delay intestinal repair and worsen the condition (Figure 3B) [56]. This phenomenon may be attributed to the immune status of the host, and the inhibition of the protective effects of IL-33 during the acute inflammatory phase could be due to the loss of functional sites caused by epithelial cell necrosis (Figure 3A,B) [56]. Hence, the contradictory effects of IL-33 seem to be contingent on the specific stage of inflammation.

Despite the debate, multiple studies have demonstrated that IL-33 plays a modulatory role in the functioning of Tregs, B cells, and innate immune cells such as macrophages and innate lymphoid cells (ILCs). Overall, the mechanism of action of IL-33 during IBD is complicated and not well understood. The various effects of IL-33 might be due to the variations in the intestinal immune micro-environment, as well as genetic and immunomodulatory effects which, in turn, can affect the effect of IL-33. This interplay increases the complexity of the effects of IL-33. Further research is needed to fully understand the mechanisms of action of IL-33 during IBD and to develop targeted interventions.

## 7. IL-33 Is Involved in Parasitic Infections

The primary immune response against parasitic infections is type 2 immunity [116], which is characterized by the infiltration of mast cells, eosinophils, and basophils, and the release of cytokines such as IL-13, IL-5, IL-4, and IL-25 [40]. ILC2 can secrete IL-13 to promote mucus secretion by goblet cells, thus playing a crucial role in parasitic infections; this process is reliant on IL-33 [117]. Interestingly, this evidence was further confirmed using mast cells from Spib^−/−^ mice [40]. In a novel discovery, IL-33 was found to promote the secretion of serotonin (5-HT) by enterochromaffin (EC) cells in the intestines (Figure 2) independently of the MyD88 signaling pathway. Chen’s experiments demonstrated that the IL-33/ST2 signaling pathway preferentially activates the TRPA1 channel in EC cells through PLC-γ1, leading to an influx of Ca^2+^ and the rapid release of a large amount of 5-HT by EC cells. This promotes intestinal peristalsis and parasite expulsion, which is a mechanism distinct from the previously identified involvement in the modulation of intestinal immunity via type 2 immunity [118].

Following its release into the extracellular space, mature IL-33 (reduced form of IL-33) can undergo oxidation and form intramolecular disulfide bonds, leading to loss of activity (oxidized IL-33) [104]. Interestingly, recent studies have found that secretions during parasitic infections can inhibit the occurrence of allergic reactions [119]. Since then, studies have shown that mice infected with *H. polygyrus* secrete proteins such as HpARI (*H. polygyrus* Alarmin Release Inhibitor), HpBARI (*H. polygyrus* Binds Alarmin Receptor and Inhibits), and Hp-TGM. Of these, HpARI can bind to the nuclear DNA of necrotic cells, thereby inhibiting the release of reduced IL-33 and its binding to the ST2 receptor (Table 2). It is worth noting that HpBARI_Hom2, a homolog of HpBARI, can bind to human ST2, suppressing allergic reactions induced by *Alternaria* (Table 2) [88,120]. Hp-TGM is a TGF-β-like molecule that can exert TGF-β effects [121]. The discovery of these novel small molecules and their functions may offer new therapeutic approaches against IL-33-mediated pathologic responses.

## 8. The Role of IL-33 in Gastrointestinal Cancer

IL-33 is involved in the development of various tumors [122]. Its expression is particularly upregulated in colorectal and gastric cancer patients, where it is primarily secreted by tumor epithelial cells, whereas its receptor, ST2L, is predominantly expressed in intestinal epithelial stromal cells [81,123,124]. IBD has been shown to increase the risk of colorectal cancer [125,126,127]. However, the underlying involvement of IL-33 in tumors is unknown. The majority of research implies that IL-33 promotes tumor migration, proliferation, and angiogenesis in gut cancer [128,129,130]. Nevertheless, studies have also shown that IL-33 has anti-tumor properties [81,131]. Extensive research is underway to explore various targeted therapeutic approaches aimed at modulating the effects of IL-33 in gastrointestinal cancer.

IL-33 can activate subepithelial myofibroblasts (SEMFs) and mast cells prior to the beginning of carcinoma of the colon, leading to the production of Th2 cytokines. Additionally, it stimulates tumor development by inducing the secretion of several growth factors such as FGF and AREG and pro-angiogenic factors like VEGFC, and producing an immune milieu biased towards Th2 responses (Figure 3B) [124]. IL-33 activates tumor-associated macrophages and dendritic cells by binding to its receptor, ST2, triggering the release of abundant Th2 cytokines, including IL-10, IL-4, and IL-13, which inhibit immune cell responses, hampering tumor clearance and control. The evidence, therefore, indicates that blocking the IL-33/ST2 axis can inhibit tumor development (Figure 3C) [130]. IL-11, which is derived from tumor cells, is a crucial factor involved in inducing IL-33 production in gastric cancer cells (Figure 3C) [132]. Recent studies show that increased IL-33 expression in colon cancer cells promotes PPARγ expression in ILC2s, leading to IL-13 release and facilitating tumor migration (Figure 3C). Inhibiting PPARγ suppresses ILC2s’ pro-metastatic effects, and hence offers promise as a target in cancer therapy [72]. Furthermore, the absence of mast cells inhibits intestinal-type gastric tumor growth, possibly due to IL-33’s influence on mast cells, hence promoting factors associated with macrophage growth, recruitment, and angiogenesis and supporting tumor growth [132]. Another study demonstrated the accumulation of Treg cells, which possess immunosuppressive properties, within the tumor, thus hindering tumor elimination. However, when Treg cells lack IL-33, the tumor microenvironment changes, causing the cells to transform into “fragile” Treg cells with considerably increased PD-1 expression, lower neuropilin-1 (Nrp-1) expression, and loss of IL-33’s inhibitory effect on NF-κB. This leads to activation of the NF-κB-T-bet axis, promoting IFN-γ production and subsequently facilitating tumor regression [47]. Furthermore, inhibiting the IL-33/ST2 pathway in a mouse model of AOM/DSS-induced colorectal cancer can reduce the proliferation and migration of CD4^+^Foxp3^+^ST2^+^ Treg cells by downregulating integrins and chemokine receptor expression. Additionally, it can inhibit the differentiation of Th17 cells and the expression of IL-17A while promoting the cytotoxic activity of CD8^+^ cells, thus aiding in tumor regression [133]. Compared with healthy cecum, the orthotopic cecal wall tumor model has ST2^+^ TAMs (tumor-associated macrophages) that, under the influence of IL-33, limit M2 polarization and the cytotoxic activity of CD8^+^ T cells (Figure 3C) [130]. Apart from its indirect effects, IL-33 also directly promotes colorectal cancer. IL-33 binds to the ST2 receptor, triggering the activation of NF-κB, which upregulates the expression of COX2 and PGE2 production, thereby promoting CRC growth (Figure 3C) [123].

The upstream regulation of IL-33 is currently receiving attention. NFE2L3, as a transcription factor, can promote cell proliferation by alleviating DUX4-mediated inhibition of CDK1. NFE2L3^−/−^ can inhibit the development of colorectal cancer [134]. Saliba’s experiment demonstrated that NFE2L3 also partially contributes to the growth and spread of IBD-associated CRC by increasing the expression of IL-33 and RAB27A/B in mast cells. NFE2L3 deficiency leads to decreased expression of IL-33 in mast cells, activation of the RAB signaling pathway, and the inhibitory effect of Treg cells, all of which slow the progression of colitis and IBD-related colon cancer [135].

The differing conclusions on the role of Treg cells in the mentioned experiments may be attributed to their heterogeneity in different immune microenvironments. Some studies suggest that Treg cells can suppress inflammation and inhibit inflammation-induced tumorigenesis. However, their presence after tumor development may promote tumor growth by inhibiting the cytotoxic activity of CD8+ T cells (Figure 3B,C) [136]. Mast cells, whose proliferation depends on IL-10, also exhibit this effect [129,137]. This indicates the importance of considering the immunological cell’s contrasting effects at distinct stages.

IL-33, in contrast to earlier findings, has been observed to induce CCL2 expression in colorectal cancer cells, leading to the recruitment of macrophages and exertion of anti-tumor effects. ST2L^−/−^ from tumor cells can promote the growth of cancer cells [131]. Moritz’s study backs up the evidence, showing that IL-33 can activate the non-hematopoietic NF-κB pathway in a mouse model of sporadic colon cancer, leading to increased expression of IFN-γ, CXCL10, and CXCL11, thereby reducing the incidence of intestinal tumors [81].

The above experimental results demonstrate the function of IL-33 in the upregulation and downregulation of pathways in colorectal cancer, indicate the complexity of the role of IL-33, and provide a novel concept for further improving the function of IL-33 derivatives for tumor control.

## 9. Therapeutic Drugs and Targets for IL-33-Related Diseases

IL-33 is closely associated with a variety of diseases, and the regulatory mechanisms centered on IL-33 are now being explored. Certain small molecules such as miR-487b [138] and SPRR3 [139] can suppress the IL-33/ST2 signaling pathway, relieving chronic heart failure and allergic asthma, respectively (Table 1). IL-37D, on the other hand, can upregulate the IL-33/ST2 signaling pathway (Table 1), participating in the regulation of obesity and metabolic disorders [140]. The p38α/β MAPK signaling pathway [141] and the Notch1 signaling pathway [142] are also associated with the response and expression of IL-33 in ILC2 and B cells (Table 1). These molecules and signaling pathways might be targets in the IL-33/ST2 axis.

In parallel, certain drugs and small molecules have been identified as promising treatment approaches for IL-33-related diseases. Resveratrol has been shown to target the MK2/3-PI3K/Akt axis in allergic diseases, inhibiting IL-33-mediated mast cell activation and therefore alleviating symptoms (Table 2) [143]. Notably, Ercolano G., found that blocking PPARγ in IL-33-dependent tumors such as CRC can greatly suppress the IL-33-stimulated release of type 2 cytokines from ILC2, thus decreasing tumor migration (Table 2) [72]. Several other inhibitors of the IL-33/ST2 axis, including Honokiol [144], Tranilast [145], and EF24 [100] and activators such as Flagellin of AIEC [146], HpARI_CCP1/2 [147], and Statins [148] have been identified as potential therapeutic agents for IL-33-related diseases (Table 2).

In summary, the meticulous exploration of these intricate mechanisms and drugs proffers a constructive avenue providing a unique, forward-thinking viewpoint for the potential therapeutic therapy of IL-33-related diseases.

## 10. Conclusions

IL-33 has complex functions in IBD and gastrointestinal cancer, with varying impacts across various immunological microenvironments. Therefore, while analyzing the function of IL-33, it is critical to evaluate its functional distinction in various settings in order to investigate its particular modes of action. IL-33 has a substantial impact on inflammatory autoimmune disorders, gastrointestinal cancer, and parasitic infections, and regulating its signaling pathways may aid in the treatment of various disorders. The major therapy options are interventional treatments using new small molecules and activation or inhibition of critical factors.

Increasing evidence shows that, in addition to its involvement in type 2 and type 1 immunity, IL-33 affects a variety of physiological processes that are dependent on the microenvironment and the distribution of the ST2 receptor on target cells. In addition to its alarm function, IL-33 also serves as a pleiotropic cytokine. However, many elements of IL-33 biology remain unknown, as do certain fundamental biological problems. In particular, how is IL-33 mRNA and protein expression controlled under healthy and pathological conditions? As our understanding of the influence of IL-33 on many organs in chronic inflammation and cancer diseases develops, the increased interest in IL-33 throughout the scientific community should lead to the development of speedy solutions to these problems.

## Figures and Tables

**Figure 1 ijms-24-13504-f001:**
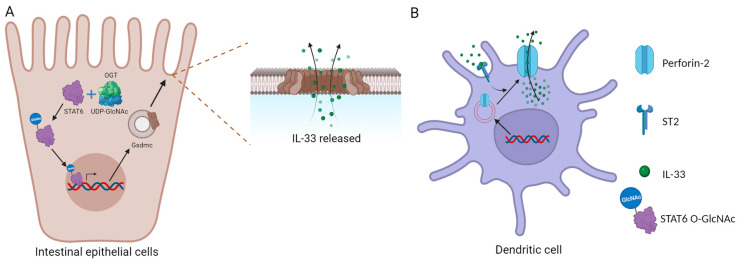
Specific IL-33 release mechanisms. (**A**) IL-33 released through the Gsdmc pore; (**B**) IL-33 released through perforin-2.

**Figure 2 ijms-24-13504-f002:**
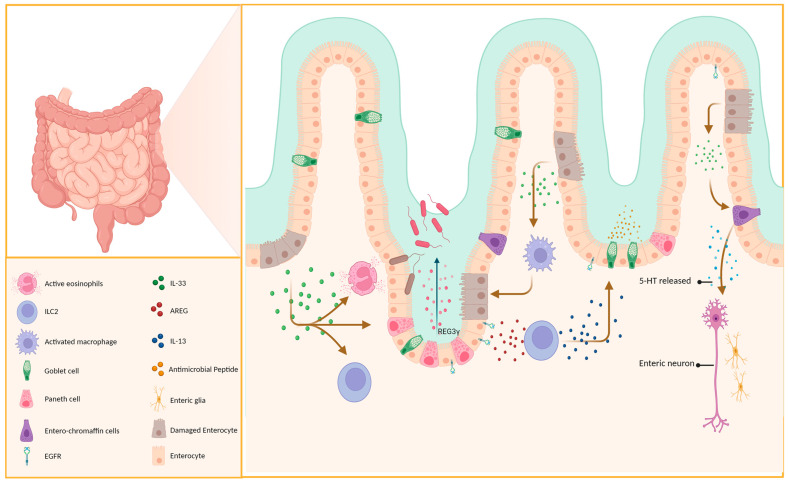
IL-33 has a protective effect on epithelial cells. Paneth cells and intestinal enterochromaffin cells secrete immune-active substances. This helps maintain the integrity of the intestinal epithelial cells and prevents pathogen invasion.

**Figure 3 ijms-24-13504-f003:**
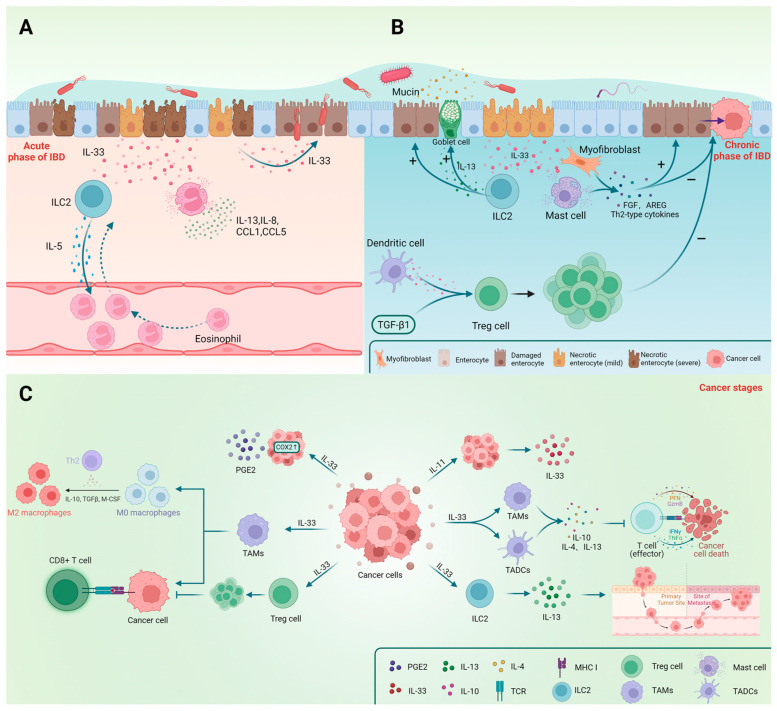
The main effects of IL-33 and immune cells in the three stages of IBD (acute phase, chronic phase, and cancer) are shown in the figure. IL-33 and ILC2 play crucial roles in the repair and protection of epithelial cells during the chronic and recovery phases. The effects of Treg cells and mast cells differ before and after cancer occurrence. (**A**) IL-33 plays a role in the acute phase of IBD; (**B**) The involvement of IL-33 during the chronic stage of IBD; (**C**) The impact of IL-33 on gastrointestinal cancer.

**Table 1 ijms-24-13504-t001:** Potential IL-33/ST2 pathway-related therapeutic targets.

	Related Diseases or Cells	Mechanism	Refs.
miR-487b	Chronic heart failure	Inhibits the expression of IL-33 and ST2.	[138]
IL-37D	Obesity and metabolic disorders	Inhibits the production of sST2 in white adipose tissue (WAT) via the IL-1R8 receptor, upregulating IL-33/ST2 signaling.	[140]
p38α/β MAPK pathway	ILC2	Inhibits the p38α/β MAPK pathway, reduces the sensitivity of ILC2 to IL-33, and inhibits the production of type 2 cytokines.	[141]
Notch1 signaling pathway	B cells	Upregulates the expression of IL-33 in B cells, hence regulating T cell immune response.	[142]
Focal adhesion kinase (FAK)	Squamous cell carcinoma cells (SCCs)	FAK upregulates the expression of IL-33 and sST2 via RUNX1 and SP1.	[50]
SPRR3	Allergic asthma	Reduces the expression of IL-33, attenuating the activation of the PI3K/AKT/NF-κB signaling pathway in ILC2, and inhibiting allergic airway inflammation.	[139]

**Table 2 ijms-24-13504-t002:** Potential therapeutic drugs or small molecules for IL-33-related diseases.

	Function	Related Diseases or Cells	Mechanism	Refs.
Resveratrol	Inhibitor	Allergic disease	Inhibits IL-33/ST2-mediated mast cell activation by targeting the MK2/3-PI3K/Akt pathway downstream of p38 instead of the NF-κB pathway.	[143]
Honokiol	Inhibitor	Lupus nephritis (LN)	Downregulates the expression of NLRP3 and ST2 to inhibit abnormal interaction between renal tubular epithelial cells and renal macrophages.	[144]
HpBARI_Hom2	Inhibitor	?	Binds, with high affinity, to human ST2 to inhibit the binding of IL-33 to ST2.	[120]
HpARI	Inhibitor	?	Binds to IL-33 and prevents the binding of active IL-33 to ST2.	[88]
HpARI_CCP1/2	Activator	?	Capable of stabilizing IL-33, increasing its half-life, and amplifying its effects.	[147]
Tranilast	Inhibitor	Allergic disease	Tranilast significantly inhibits LPS-induced Akt activation and downregulates the expression of IL-33.	[145]
Statins	Activator	Cardiovascular diseases	Inhibits the mevalonate (MAV) pathway, suppressing the geranylgeranylation of RhoA, leading to the inhibition of ROCK and the inactivation of SRF, thereby relieving SRF-mediated transcriptional repression of IL-33 gene expression and upregulating IL-33 expression.	[148]
Flagellin of AIEC	Activator	Intestinal fibrosis	Promotes the expression of ST2 in IEC, activates the IL-33-ST2 pathway, and enhances intestinal fibrosis in the context of colitis.	[146]
EF24	Inhibitor	?	Upregulates the expression of ST2 and SIGIRR, reduces the interaction between ST2 and MyD88, and significantly downregulates the expression of IL-33 in LPS-stimulated DCs.	[100]
PPARγ	Activator	Colorectal Cancer	Promotes the secretion of IL-13 by ILC2s, thereby facilitating tumor migration and invasion.	[72]

## Data Availability

Not applicable.

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
