# Peer review of "A Deep View of the Biological Property of Interleukin-33 and Its Dysfunction in the Gut"

_ijms, 2023, doi:10.3390/ijms241713504_

Round 1
Reviewer 1 Report
The article is consistent within itself. The references are relevant and recent. The cited sources are referenced correctly. Appropriate and key studies are included. The paper is comprehensive, the flow is logical and the data is presented critically.
However, there are some specific comments on weaknesses of the article and what could be improved:
Major points - none
Minor points
1. The biological characteristics of IL-33 should be described in the light of gut, because this is the focus of the paper. Please, add some discussion in the text. Otherwise, it`s too extensive and non-related.
2. "5. The pathogenesis of IBD" - please, precise the heading accordingly and connect with the next section.
3. Table 1 is not mentioned in the text. The same goes for table 2. Probably, you should have a section on the IL-33 as a potential therapeutic molecule?
The English is good.
Author Response
Reviewer 1 comments
1.The biological characteristics of IL-33 should be described in the light of gut, because this is the focus of the paper. Please, add some discussion in the text. Otherwise, it`s too extensive and non-related
Response:Thank you very much for pointing out the limitation in our article. Based on your suggestion, we added a discussion of “The Role of IL-33 in the Intestine” on page 7 and 8 line 313-326, and have marked it in bright blue. Thank you again for the warm reminder
2."5. The pathogenesis of IBD" - please, precise the heading accordingly and connect with the next section.
Response:We greatly appreciate your valuable feedback in identifying the issues within our article. We have made a modification to the heading upon your suggestion. Please refer to page 9, line 393. Thank you again for your help.
- Table 1 is not mentioned in the text. The same goes for table 2. Probably, you should have a section on the IL-33 as a potential therapeutic molecule?
Response:We greatly appreciate your insights highlighting the issues in our article. In accordance with your suggestions, we have added the section titled "Therapeutic drugs and targets for IL-33-related diseases" on pages 13 and 14, lines 598-620 of the article. Additionally, we have provided annotations for Tables 1 and 2 within this section.Thank you again for your help.

Reviewer 2 Report
The authors have discussed about the biological property of the pro-inflammatory cytokine, IL-33 and its dysfunction in the intestinal microenvironment.
Overall, the review is well structured and is communicable to the general audience.
However, authors can address few minor issues to enhance the quality of the review article.
1. Suggest writing Interleukin-33 in the beginning line and abbreviating it as IL-33 (line number 42).
2. Suggest writing the full name of Gsdmc at least once in the article.
3. The authors should increase the font of the figure labels wherever necessary for better visibility (Figure 1, 2 and 3).
4. Check the wrong figure number (line number 583).

Author Response
Reviewer 2 comments
1.Suggest writing Interleukin-33 in the beginning line and abbreviating it as IL-33 (line number 42).
Response:Thank you very much for pointing out the shortcomings in our article. According to your generous suggestion, we have rewritten “Interleukin-33” in the beginning line and abbreviating it as IL-33 on page 2 and line 42, All of these changes have been highlighted in bright blue. We express our gratitude once more for your assistance and guidance.
2.Suggest writing the full name of Gsdmc at least once in the article
Response:We deeply value your perspectives that have illuminated the issues within our article. Following your recommendations, we have included the full name of Gsdmc at line 33 on the first page of the article using a bright blue color. We extend our thanks once again for your valuable support and direction.
3.The authors should increase the font of the figure labels wherever necessary for better visibility (Figure 1, 2 and 3)
Response:We extend our sincere gratitude for highlighting the issues within our article. Due to your suggestion, we have added the font of the figure labels in pursuit of enhanced visibility of Figures 1, 2, and 3. Once again, we appreciate your guidance.
4.Check the wrong figure number (line number 583)
Response:Thank you very much for pointing out the fault in our article.Following your nice recommendations, we have corrected the wrong figure number on page 13 and line 593 using a bright blue color.Thanks again.
